# Exploiting Machine Learning Technologies to Study the Compound Effects of Serum Creatinine and Electrolytes on the Risk of Acute Kidney Injury in Intensive Care Units

**DOI:** 10.3390/diagnostics13152551

**Published:** 2023-07-31

**Authors:** Hsin-Hung Liu, Yu-Tseng Wang, Meng-Han Yang, Wei-Shu Kevin Lin, Yen-Jen Oyang

**Affiliations:** 1Graduate Institute of Biomedical Electronics and Bioinformatics, National Taiwan University, Taipei City 10617, Taiwan; qi3800@yahoo.com.tw; 2Graduate Institute of Networking and Multimedia, National Taiwan University, Taipei City 10617, Taiwan; d05944001@ntu.edu.tw; 3Department of Computer Science and Information Engineering, National Kaohsiung University of Science and Technology, Kaohsiung City 807618, Taiwan; menghanyang@nkust.edu.tw; 4Department of Emergency Medicine, National Taiwan University Hospital, Taipei City 10002, Taiwan; 5Department of Computer Science and Information Engineering, National Taiwan University, Taipei City 10617, Taiwan

**Keywords:** acute kidney injury, serum electrolyte, intensive care unit, machine learning

## Abstract

Assessing the risk of acute kidney injury (AKI) has been a challenging issue for clinicians in intensive care units (ICUs). In recent years, a number of studies have been conducted to investigate the associations between several serum electrolytes and AKI. Nevertheless, the compound effects of serum creatinine, blood urea nitrogen (BUN), and clinically relevant serum electrolytes have yet to be comprehensively investigated. Accordingly, we initiated this study aiming to develop machine learning models that illustrate how these factors interact with each other. In particular, we focused on ICU patients without a prior history of AKI or AKI-related comorbidities. With this practice, we were able to examine the associations between the levels of serum electrolytes and renal function in a more controlled manner. Our analyses revealed that the levels of serum creatinine, chloride, and magnesium were the three major factors to be monitored for this group of patients. In summary, our results can provide valuable insights for developing early intervention and effective management strategies as well as crucial clues for future investigations of the pathophysiological mechanisms that are involved. In future studies, subgroup analyses based on different causes of AKI should be conducted to further enhance our understanding of AKI.

## 1. Introduction

Acute kidney injury (AKI) is a condition frequently encountered in medical care [1]. The underlying pathophysiological processes of AKI ultimately lead to a decline in renal function. As a result, the patients suffer from the accumulation of waste products, an imbalance of electrolytes, and a widespread inflammatory response that affects organs beyond the kidneys [2]. According to a recent study, 20% to 50% of the patients in an intensive care unit (ICU) suffered from AKI [3]. Therefore, how to assess the risk of AKI is a critical issue for clinicians in an ICU [4]. However, several early signs of AKI, including edema, hypertension, and oliguria, are non-specific. Therefore, the current practice only monitors the level of serum creatinine and the volume of urine output in order to assess the risk of AKI [5,6].

Due to the observation above, scientists have been investigating the physiological signs that may be associated with the development of AKI. Leaf et al. conducted a review of the pathophysiology of dysregulated mineral metabolism, specifically focusing on calcium, phosphate, parathyroid hormone, and vitamin D metabolites in the context of AKI [7]. A review conducted by Yokota et al. found that the most common comorbidities associated with AKI in elderly patients included respiratory failure, cardiovascular disease, hypertension, diabetes, surgical complications, and liver disease [8].

As the kidneys play a crucial role in regulating the balance of calcium, phosphate, and magnesium, it is conceivable that an imbalance of serum electrolytes may be associated with the development of AKI. In this respect, a previous study reported that acute phosphate nephropathy was an early condition of AKI and might subsequently progress to chronic renal failure [9]. Furthermore, a number of studies were conducted to investigate how the levels of serum electrolytes, including chloride, phosphorus, magnesium, potassium, sodium, and calcium, were associated with the development of AKI [10,11,12]. Suetrong et al. observed a linear correlation between the concentration of serum chloride and the development of AKI among sepsis/septic shock patients [13]. Marttinen et al. reported a similar result and showed that the temporal chloride level was associated with an increased risk of AKI [14]. The work by Moon et al. revealed that a high level of serum phosphorus increased the risk of AKI [15]. Cheungpasitporn et al. showed that both hypomagnesemia and hypermagnesemia led to an increased risk of in-hospital AKI [16]. Thongprayoon et al. observed a U-shaped association between the level of serum ionized calcium and in-hospital AKI. Furthermore, both hypocalcemia and hypercalcemia were reported to be associated with an increased risk of hospital-acquired AKI [17,18], and Chen et al. discovered that abnormal levels of serum sodium or potassium before an AKI diagnosis were more likely to lead to AKI progression and a poor prognosis [19]. Nevertheless, Yessayan et al. reported that the concentration of hyperchloremia and the onset of AKI within 72 h of admission were not correlated [20]. Finally, Morooka et al. divided pediatric patients into three groups based on their serum magnesium values and investigated the association between magnesium levels and outcomes [21].

In addition to the studies addressed above, the latest trend is to exploit various machine learning algorithms, including artificial neural networks [22], support vector machines (SVMs) [23], Bayesian networks [24], random forests (RFs) [25], etc., to predict incidences of AKI, and Song et al. reviewed how the conventional logistic regression (LR) and various machine learning methods performed in this respect [26]. A representative study was conducted by Tomasev et al. [27]. In their study, the authors employed a recurrent neural network to build their prediction models based on a cohort of 703,782 cases collected from the medical facilities of the U.S. Department of Veterans Affairs.

Though the effects of several serum electrolytes on the development of AKI have been well reported, a comprehensive investigation into how these serum electrolytes interact in the context of the development of AKI has not been conducted [28,29,30,31,32,33,34]. It is conceivable that such studies can provide crucial clues for developing new clinical guidelines to assess the risk of AKI. Accordingly, we initiated this study aiming not only to illustrate how these factors interact with each other but also to provide new insights for developing new clinical practices. Our analyses focused on ICU patients who had no prior history of AKI and were free of AKI-related comorbidities such as diabetes and hypertension as well as common causes of AKI such as hypovolemia and heart failure. By focusing on this group of patients, we were able to eliminate the confounding influences of these conditions and examine the associations between the levels of serum electrolytes and renal function in a more controlled manner.

In this study, we exploited decision tree (DT) models [28,29,30] and RF models [31,32]. Compared to the other commonly exploited machine learning models, such as SVMs [23] and deep neural networks (DNNs) [22], DT and RF models are favorites in many applications due to the interpretable decision rules exhibited by these models. Figure 1 shows a DT structure that summarizes the main results of this study. A user can figure out the decision rules by traversing the tree structure from the root node, which is at the top of the structure and colored yellow. They can proceed by following the branch originating from the root node that matches the condition of the case. The path ends at one of the leaf nodes at the bottom level of the tree. The “n^+^” and “n^−^” symbols at each node denote the number of positive cases and the number of negative cases, respectively, in our study cohort that met the criteria specified along the path from the root node to this particular node. If a path ends at a red node, the prediction is positive. On the other hand, if a path ends at a green node, the prediction is negative. Based on these interpretable decision rules, physicians can have a comprehensive understanding of how these key factors interact with each other and develop new clinical guidelines accordingly. On the other hand, due to the non-linear transformations and the large number of coefficients involved in the prediction process, it is essentially impossible for a user to interpret the mathematics equations that an SVM or DNN model follows to make a prediction.

## 2. Materials and Methods

### 2.1. Study Cohort

Our study cohort was extracted from the Medical Information Mart for Intensive Care (MIMIC)-IV database, version 1.0, published in March 2021 [33,34]. The MIMIC database has been carefully de-identified to protect patient privacy. Its use for research purposes has been approved by the institutional review boards of the Massachusetts Institute of Technology (Protocol No. 0403000206) and Beth Israel Deaconess Medical Center (Protocol No. 2001-P-001699/14). These approvals indicate that the appropriate ethical considerations have been taken into account to ensure the responsible and lawful use of the database for research purposes.

Figure 2 shows the flow that we followed to generate our study cohort. Initially, the dataset contained 256,878 clinical records collected at the emergency department and the intensive care unit between 2008 and 2019. According to the 2012 Kidney Disease: Improving Global Outcomes (KDIGO) recommendation statements [35,36,37,38], AKI is defined by any of the following criteria: (1) an increase in the level of serum creatinine by 0.3 mg/dL (26.5 µmol/L) or more within 48 h or (2) an increase in the level of serum creatinine to 1.5 times the baseline level within 7 days. As the guideline requires two readings of the serum creatinine level and our study focused on patients in ICUs, 205,482 records in the database were excluded due to a lack of required information after admission into ICUs. As a result, only 51,396 records, all of which corresponded to the first available data after ICU admission, were included for subsequent analyses.

Since one patient could be admitted into the ICU more than one time, for a patient who had suffered from AKI, we included only the record corresponding to his/her stay in the ICU during which the patient suffered from AKI the first time. On the other hand, for a patient who had never suffered from AKI, we included only the record corresponding to his/her first stay in the ICU. As a result, only 41,878 records corresponding to 41,878 individual cases remained. In the next step, we employed the criteria provided in Table 1 to exclude those patients whose medical records showed AKI-related comorbidities [39] so that the interferences from other factors such as renal impairment, cardiac failure, diabetes, and electrolyte imbalances would be avoided. After this step, only 17,085 cases remained in the dataset. Finally, we employed the following excluding criteria to further screen the dataset: (1) the record of the case did not include all the readings listed in Table 2; (2) one or more readings in the record were in the highest 0.1% or the lowest 0.1% of the distributions; and (3) one or more readings for the case were not made within 168 h of admission. In the end, our study cohort contained 550 AKI-positive cases and 12,152 AKI-negative cases. A demographic analysis of the study cohort is presented in Table 2.

Etiologically, the causes of AKI can be classified into three broad categories: pre-renal azotemia, intrinsic renal parenchymal damage, and post-renal obstruction. Tailoring treatment plans according to the specific causes of renal injury are crucial for improving patient outcomes. For instance, hypovolemia, often diagnosed by assessing a fluid status imbalance, insufficient renal perfusion, or inferior vena cava collapse, is a common clinical presentation associated with pre-renal azotemia. On the other hand, post-renal injury occurs when the urinary tract is partially or completely blocked due to functional or structural derangements anywhere from the renal pelvis to the tip of the urethra. Since the treatment plans for post-renal AKI patients are significantly different from the plans for non-post-renal AKI patients, we classified the AKI patients in our study cohort into two categories: post-renal AKI and non-post-renal AKI. According to several previous studies, the incidences of post-renal AKI accounted for less than 5% of all AKI cases [1,40,41]. In our study cohort, 24 out of 550 AKI cases, i.e., 4.4%, were post-renal, and the percentage was in line with the previous studies. Appendix A shows the ICD-9 and ICD-10 codes employed to identify post-renal AKI cases. Table 3 shows the statistics of the post-renal AKI patents and non-post-renal AKI patients with respect to the features listed in Table 2.

**Table 2 diagnostics-13-02551-t002:** Demographic analysis of the study cohort.

Feature	550 Cases with AKI (Mean ± SD)	12,152 Cases without AKI (Mean ± SD)	*p*-Value
Age (years)	65.68 ± 14.69	60.34 ± 17.67	*p* < 0.001 *
Gender			*p* < 0.001 *
Male (%)	349 (63.45%)	6757 (55.60%)	
Female (%)	201 (36.55%)	5395 (44.40%)	
Serum			
BUN (mg/dL)	26.74 ± 15.39	18.06 ± 8.90	*p* < 0.001 *
Creatinine (mg/dL)	1.36 ± 0.64	0.86 ± 0.26	*p* < 0.001 *
Chloride (mEq/L)	110.37 ± 6.60	107.39 ± 5.28	*p* < 0.001 *
Potassium (mEq/L)	4.79 ± 0.75	4.47 ± 0.63	*p* < 0.001 *
Sodium (mEq/L)	142.81 ± 5.77	141.23 ± 4.59	*p* < 0.001 *
Magnesium (mg/dL)	2.53 ± 0.52	2.28 ± 0.44	*p* < 0.001 *
Phosphorus (mg/dL)	4.40 ± 1.34	3.80 ± 0.93	*p* < 0.001 *
Non-ionized calcium (mg/dL)	8.76 ± 0.73	8.73 ± 0.71	0.346

The symbol * indicates statistical significance. For categorical variables, the *p*-values were calculated based on the χ^2^ test [42,43]. For continuous variables, the *p*-values were calculated based on the *t*-test [42,43]. SD represents standard deviation.

### 2.2. Machine Learning Models

As mentioned earlier, we used DT and RF models in order to investigate the compound impacts of two or more factors and provide a clear picture of how these factors interact with each other. In particular, we focused on the compound effects of serum creatinine, BUN, and the 6 serum electrolytes listed in Table 2. The serum creatinine and BUN were included because in medical practice the levels of serum creatinine and BUN as well as the BUN-to-creatinine ratio are measured to clarify different types of renal function impairment, including pre-renal azotemia, intrinsic renal parenchymal disease, and post-renal obstruction. The 6 serum electrolytes listed in Table 2 were included because previous studies had reported their associations with the development of AKI.

In order to address the needs in different clinical scenarios, we generated prediction models with varying levels of sensitivity and examined the prediction rules embedded in these models. In this respect, we set the parameters of the machine learning packages to various combinations and then employed a 5-fold cross-validation [22] to evaluate the levels of sensitivity delivered by the prediction models generated with these alternative parameter settings. In the 5-fold cross-validation process, the study cohort was randomly and evenly partitioned into 5 subsets. For each combination of parameter settings, every subset was employed to evaluate the prediction models generated with the other 4 subsets. Then, the evaluation results of these 5 subsets were collected to calculate the performance data, i.e., the sensitivity, specificity, positive predictive value (PPV), etc., corresponding to this particular parameter combination. Appendix A shows the software packages employed to generate the DT and RF models as well as the alternative parameter settings employed to generate the prediction models in the 5-fold cross-validation process. In this respect, we tried a large number of possible parameter combinations in order to generate prediction models that delivered sensitivity at the levels of 0.95 and 0.80. Furthermore, as we had only 550 positive cases in our study cohort, we employed the 5-fold cross-validation process instead of the 10-fold cross-validation process, which may be more commonly used in machine learning research, so that each partition would contain a good number of positive cases.

## 3. Results

As mentioned above, in order to address the needs in different clinical scenarios, we generated prediction models with varying levels of sensitivity. In the subsequent discussions, we will focus on the prediction models with sensitivity at the levels of 0.95 and 0.80. Table 4 summarizes the performances of the DT, RF, and LR models observed during the 5-fold cross-validation procedure. The performances of the LR models were included to provide a reference because LR models are widely employed in biomedical research communities. Detailed performance data are presented in Appendix A.

The performance data in Table 4 reveal that with respect to the specificity, the positive predictive value (PPV), the relative risk, and the area under the receiver operating characteristic curve (AUC), the DT model that delivered sensitivity at the level of 0.95 performed significantly superior to the RF model that delivered the same level of sensitivity. It was also observed that the RF model that delivered sensitivity at the level of 0.80 performed marginally superior to the rival DT model in terms of specificity, PPV, and relative risk but performed inferior to the rival DT model in terms of AUC. Based on these observations, we concluded that the overall performance of the DT models was superior to that of the RF models. Therefore, in the subsequent discussions, we will focus on the DT models and the decision rules embedded in the models. 

Figure 3a,b show the DT models generated by feeding the entire study cohort into the decision tree package with the combinations of parameters cp and prior set to (0.005 and 0.5835) and (0.01 and 0.744), respectively. According to the 5-fold cross-validation addressed above, with cp and prior set to these two combinations, the generated DT models should deliver sensitivity at the levels of 0.80 and 0.95, respectively. One interesting observation regarding the DT model shown in Figure 3a is that the model predicts a patient with a serum creatinine level higher than 1.25 mg/dL to be at high risk. This prediction rule comes very close to the serum creatinine level of 1.3 mg/dL commonly used by physicians to determine whether a patient is at high risk of progression to AKI. It is also observed that the DT model shown in Figure 3b predicts a patient with a serum creatinine level higher than 0.95 mg/dL to be at high risk. This observation implies that 0.95 mg/dL can be employed as an alternative threshold if the physician wants to increase the sensitivity of his/her medical judgment. 

The DT model shown in Figure 3a further reveals that for a patient with a serum creatinine level between 0.95 and 1.25 mg/dL, his/her level of serum magnesium can be used as a warning sign. If the reading is higher than 2.45 mg/dL, the patient is at high risk. If not, we should further examine his/her level of serum chloride. If the patient’s level of serum chloride is over 106.5 mEq/L, the patient is at high risk. 

The blue polygons in Figure 3a,b encircle the structure shared by these two DT models. According to the shared structure, for a patient with a serum creatinine level between 0.75 and 0.95 mg/dL, we should further examine his/her levels of serum magnesium and chloride. A patient is at high risk if (1) his/her level of serum chloride is higher than 113.5 mEq/L or (2) his/her level of serum chloride is between 105.5 and 113.5 mEq/L and his/her level of serum magnesium is higher than 2.35 mg/dL. Finally, since only a very limited number of positive cases in our study cohort met the criteria defined by the lower right parts of the tree structures in Figure 3a,b, we should be able to ignore the corresponding decision rules. In summary, the structures of the two DT models shown in Figure 3 illustrate that the levels of serum creatinine, chloride, and magnesium are the three major factors associated with the development of AKI. Though the level of serum phosphorus is present in these DT models, the nodes corresponding to the level of serum phosphorus are located in the lower parts of the structures, which implies that these nodes play less significant roles in the decision rules.

## 4. Discussion

As of today, the clinical practice to assess the risk of AKI is based on the 2012 KDIGO Clinical Practice Guideline for Acute Kidney Injury, which monitors only the level of serum creatinine and the volume of urine output. Since AKI could lead to many complications and even fatality, identifying the risk factors of AKI and exploiting machine learning technologies to predict AKI incidences have attracted a lot of attention in biomedical research communities. In this respect, several serum electrolytes have been reported to be associated with the development of AKI. Nevertheless, the compound effects of serum creatinine, BUN, and clinically relevant serum electrolytes have yet to be thoroughly investigated. With this observation, we initiated this study aiming not only to illustrate how these factors interact with each other but also to provide new insights for developing new clinical practices for assessing AKI risk. In particular, we focused on ICU patients who had no prior history of AKI and were free of AKI-related comorbidities. By focusing on this specific group of patients, we were able to eliminate the confounding influences of these conditions and examine the associations between the levels of serum electrolytes and renal function in a more controlled manner. Furthermore, our results can provide valuable insights for developing early intervention and effective management strategies as well as for investigating the pathophysiology of AKI.

The performance data in Table 4 show that for those patients without a prior history of AKI or AKI-related comorbidities, the relative risks with these alternative prediction models were fairly high, ranging from 9.84 to 16.89. This implies that the group of patients predicted to be positive suffered significantly higher risk than the groups of patients predicted to be negative. However, the low PPVs suggest that there would be a large number of false positives if these prediction models were put into practical use. Nevertheless, according to the numbers shown in Figure 3a, this particular DT model, if put into practical use, should predict around 57% of the patients to be negative and deliver a sensitivity around 80%. Meanwhile, according to the numbers shown in Figure 3b, this particular DT model, if put into practical use, should predict around 51% of the patients to be negative and deliver a sensitivity around 95%. Therefore, a physician who employs the DT models developed in this study to assess the risks of AKI for his/her patients only needs to focus on about 50% of the patients, while the physician can expect this group of patients to suffer about 10 times the risk of the group of patients predicted to be at low risk.

Among the 10 variables listed in Table 2, only serum creatinine, chloride, magnesium, and phosphorus are present in the DT models shown in Figure 3a,b. It must be noted that this observation does not imply that serum potassium, sodium, and non-ionized calcium are not associated with the development of AKI. In fact, as mentioned earlier, previous studies have reported that serum potassium, sodium, and non-ionized calcium are all associated with the development of AKI. What happened must be that when building the prediction model, the DT algorithm figured out that the levels of serum chloride, magnesium, and phosphorus provided more information than the levels of serum potassium, sodium, and non-ionized calcium. The DT algorithm further figured out that the additional information provided by the levels of serum potassium, sodium, and non-ionized calcium after the levels of serum chloride, magnesium, and phosphorus had been incorporated was insignificant.

The DT models shown in Figure 3a,b identify the levels of serum creatinine, chloride, and magnesium as the three major factors associated with the development of AKI. Though the level of serum phosphorus is present in these two figures, all three nodes corresponding to the level of serum phosphorus are located in the lower levels of the structures. Furthermore, only a very limited number of positive cases in our study cohort met the criteria defined by these low-level structures. Therefore, in practice, we can ignore the role of serum phosphorus. 

Since the level of serum creatinine is one of the major factors monitored in the current clinical practice, our study suggests that for those patients without a prior history of AKI or AKI-related comorbidities, the levels of serum chloride and magnesium should be taken into consideration in order to enhance the clinical guidelines. In this respect, the current clinical guideline, which monitors only the level of serum creatinine and the volume of urine output, may lead to misdiagnoses and/or delayed treatments in some cases because the level of serum creatinine generally reflects the degree of renal damage and should be considered as a delayed indicator of AKI. Furthermore, decreased urine output is a non-specific symptom and may only be evident once the AKI has progressed. Therefore, by incorporating the assessments of the serum chloride and magnesium levels into the enhanced clinical guideline, healthcare professionals can obtain a more comprehensive understanding of a patient’s renal function and the risk of AKI. Furthermore, the numbers shown in Table 2 reveal that the distributions of the levels of serum creatinine for patients with AKI and patients without AKI must overlap to a large degree because the standard deviation of the level of serum creatinine for patients with AKI, which is 0.64, is larger than the difference between the means of these two groups of patients, which is 0.5. This implies that additional assessments must be incorporated if we would like to evaluate the risk of AKI of a patient more accurately. Finally, with respect to the decrease in urine output among AKI patients, it is a non-specific symptom and may only be evident once the AKI has progressed. Together, these observations imply that for an ICU patient without a prior history of AKI or AKI-related comorbidities, healthcare professionals can obtain a more comprehensive understanding of the patient’s renal function and risk of AKI by incorporating assessments of serum chloride and magnesium levels into the enhanced clinical guideline. Accordingly, healthcare professionals will be able to evaluate and manage treatments more precisely and ultimately prevent disease progression and deterioration.

It must be noted that our results can only be immediately applied to ICU patients without a prior history of AKI or AKI-related comorbidities. For ICU patients with AKI-related comorbidities, further studies are needed. In this respect, we can partition the patients into several groups depending on the types of comorbidities that they suffer from so that patients in the same group have similar pathophysiological mechanisms. Then, we can apply the procedure presented in this article to each group of patients in order to develop a specific prediction model for each group and identify the critical factors accordingly. 

One of the major limitations of our study is due to the different causes of AKI. As the causes of AKI are essential for physicians to develop effective treatment plans, in-depth subgroup analyses based on different categories of renal injury should be conducted to gain valuable insights into the different pathophysiological mechanisms involved and guide appropriate treatment strategies tailored to each subgroup. In this study, based on the information available in the MIMIC-IV dataset, we classified the AKI patients into two categories: post-renal and non-post-renal. The statistics in Table 3 reveal that there were no statistical differences between the levels of the eight serum ingredients for the post-renal and non-post-renal AKI patients. Therefore, our prediction models should be generally applicable to both post-renal and non-post-renal AKI patients. Nevertheless, in-depth subgroup analyses should be conducted in the future.

In addition to the limitation addressed above, this is a retrospective study based on data extracted from the MIMIC-IV database. Therefore, the results derived from this study should not be extensively applied in the decision process without taking into consideration the ethnic composition of the patients and the medical interventions that these patients may have received. Furthermore, our study was based on clinical records collected in ICUs. This implies that the patients involved had serious health conditions. The data in Table 2 also show that these patients were relatively old. Therefore, the results observed in our analyses should not be generalized to patients with different health conditions and in different age groups. Finally, our results only illustrate the associations between the investigated risk factors and the incidences of AKI. In other words, causal inferences have yet to be studied. 

## 5. Conclusions

This study has led to an in-depth understanding of the compound effects of serum creatinine, chloride, and magnesium with respect to the development of AKI in ICUs. As we focused on patients who had no prior history of AKI and were free of AKI-related comorbidities, our study provides valuable insights for developing early intervention and effective management strategies. Furthermore, this understanding provides crucial clues not only for future enhancement of clinical practices but also for future investigation of the pathophysiological mechanisms that are involved.

## Figures and Tables

**Figure 1 diagnostics-13-02551-f001:**
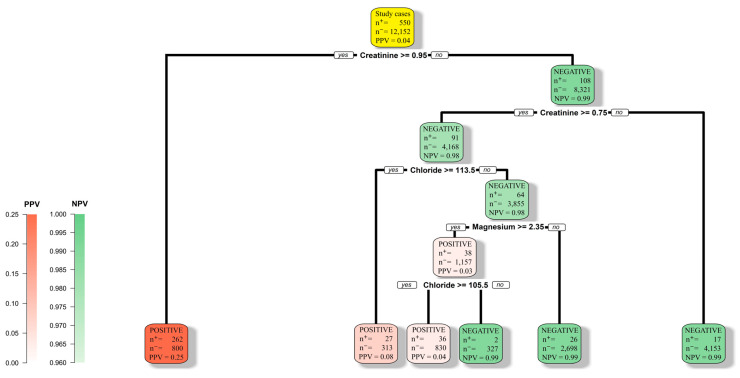
A DT structure that summarizes the main results of this study. The root node is colored yellow.

**Figure 2 diagnostics-13-02551-f002:**
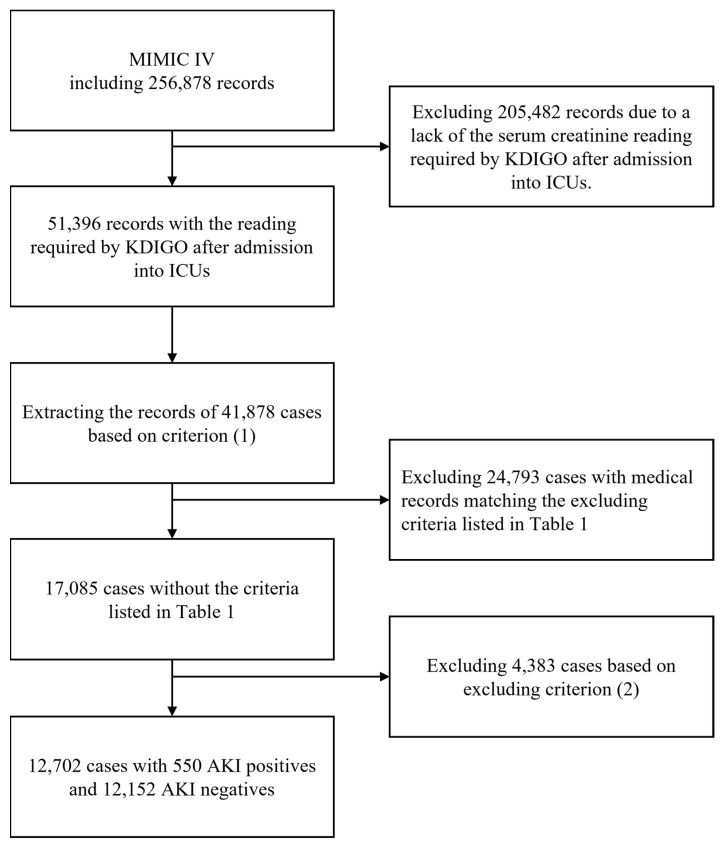
The flow for generating the study cohort. Table 1 lists the ICD-9 and ICD-10 codes employed to exclude the cases with AKI-related comorbidities/diseases. Criterion (1): (i) For a patient who had suffered from AKI, we included only the record corresponding to his/her stay in the ICU during which the patient suffered from AKI the first time. (ii) For a patient who had never suffered from AKI, we included only the record corresponding to his/her first stay in the ICU. Criterion (2): (i) the record of the case did not include all the readings listed in Table 2; (ii) one or more readings in the record were in the highest 0.1% or the lowest 0.1% of the distributions; or (iii) one or more readings in the record were not measured within 168 h of admission.

**Figure 3 diagnostics-13-02551-f003:**
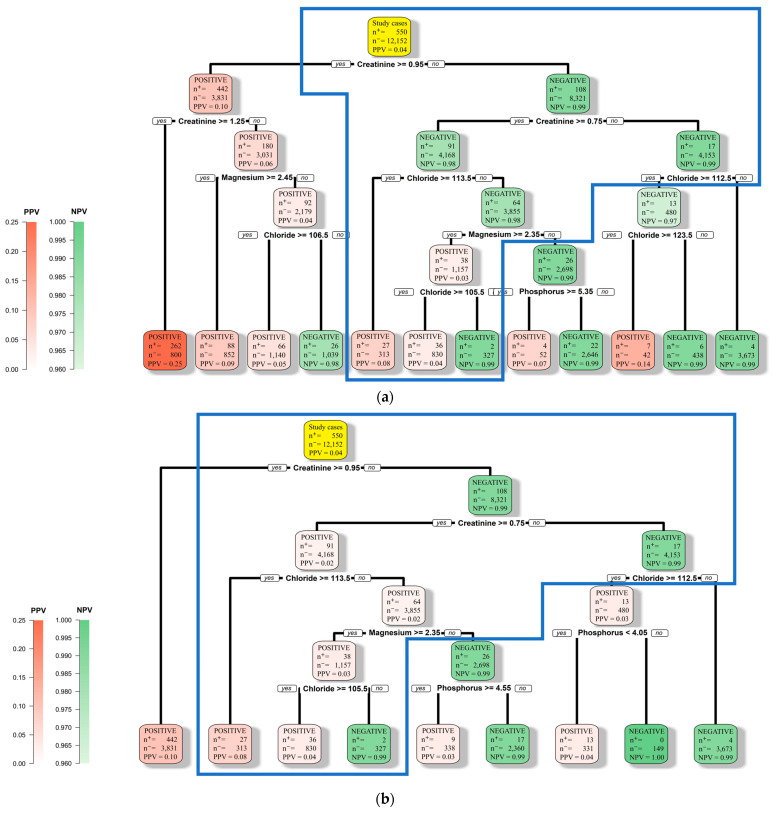
DT models with two different levels of sensitivity. (**a**) The DT model with sensitivity at the level of 0.80. (**b**) The DT model with sensitivity at the level of 0.95. The blue polygons encircle the structure shared by these 2 DT models. The root node is colored yellow.

**Table 1 diagnostics-13-02551-t001:** Excluding criteria for the cases with AKI-related comorbidities/diseases.

Comorbidities/Diseases	ICD-9	ICD-10
Renal failure ^1^	403.11, 403.91, 404.12, 404.92, 584.5–584.9, 585.1–585.9, 586, V42.0, V45.1, V56.0, V56.8	I12.0, I13.1, N17.0–N17.2, N17.8, N17.9, N18.1–N18.9, N19, N25.0, Z49.0–Z49.2, Z94.0, Z99.2
Congestive heart failure	398.91, 402.11, 402.91, 404.11, 404.13, 404.91, 404.93, 428.0–428.9	I09.9, I11.0, I13.0, I13.2, I25.5, I42.0, I42.5–I42.9, I50.0–I50.9, P29.0
Diabetes	250.0–250.7, 250.9	E10.0–E10.9, E11.0–E11.9, E12.0–E12.9, E13.0–E13.9, E14.0–E14.9
Fluid and electrolyte disorders	276.0–276.9	E22.2, E86.0, E86.1, E86.9, E87.0–E87.8

^1^ Including end-stage renal disease, AKI, and chronic kidney disease.

**Table 3 diagnostics-13-02551-t003:** Statistical analysis of the characteristics of the post-renal AKI patients and the non-post-renal AKI patients in our study cohort.

Feature	24 Cases with Post-Renal AKI (Mean ± SD)	526 Cases with Non-Post-Renal AKI (Mean ± SD)	*p*-Value
Age (years)	74.16 ± 12.54	65.30 ± 14.66	0.0037 *
Gender			0.0007 *
Male (%)	23 (95.66%)	326 (61.98%)	
Female (%)	1 (4.34%)	200 (38.02%)	
Serum			
BUN (mg/dL)	27.54 ± 10.72	26.70 ± 15.55	0.7944
Creatinine (mg/dL)	1.40 ± 0.57	1.36 ± 0.64	0.7459
Chloride (mEq/L)	110.08 ± 7.30	110.38 ± 6.57	0.5228
Potassium (mEq/L)	4.58 ± 0.56	4.80 ± 0.76	0.2258
Sodium (mEq/L)	142.81 ± 5.28	142.84 ± 5.79	0.9599
Magnesium (mg/dL)	2.54 ± 0.38	2.53 ± 0.53	0.1666
Phosphorus (mg/dL)	4.07 ± 0.93	4.41 ± 1.36	0.5254
Non-ionized calcium (mg/dL)	8.66 ± 0.54	8.76 ± 0.74	0.8278

The symbol * denotes statistical significance. The *p*-values were calculated based on the two-sample *t*-test. SD stands for standard deviation.

**Table 4 diagnostics-13-02551-t004:** Summary of the performances observed during the 5-fold cross-validation process.

Level of Sensitivity	Model	Sensitivity	Specificity	PPV	AUC	Relative Risk
0.95	DT	0.949	0.479	0.076	0.767	16.893
	LR	0.949	0.414	0.068	0.855	13.872
	RF	0.949	0.382	0.065	0.666	13.012
0.80	DT	0.798	0.721	0.116	0.823	9.84
	LR	0.799	0.773	0.137	0.857	11.982
	RF	0.799	0.732	0.119	0.766	10.141

PPV stands for positive predictive value, also known as precision. AUC stands for the area under the receiver operating characteristic curve.

## Data Availability

MIMIC-IV ver. 1.0 was provided by physionet.

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
