# Peer review of "Exploiting Machine Learning Technologies to Study the Compound Effects of Serum Creatinine and Electrolytes on the Risk of Acute Kidney Injury in Intensive Care Units"

_diagnostics, 2023, doi:10.3390/diagnostics13152551_

Round 1

Reviewer 1 Report

The authors conducted an investigation on the association of important clinical factors with Acute Kidney Injury (AKI) development through a machine learning approach. The attempt to explore this research area is both significant and intriguing. Nevertheless, a critical concern arises from the ambiguity in defining the outcome. Additionally, the paper lacks a comprehensive discussion, which is essential for strengthening the overall analysis.

1. In the abstract, the background description is overly extended, while the method and result sections lack sufficient detail. To enhance clarity and conciseness, a reconstructed and rewritten abstract is suggested.

2. Although the authors referred to the criteria, they never mentioned the causes of AKI. The causes of acute kidney injury are traditionally grouped into three categories: prerenal, renal (with direct intrinsic kidney damage), and postrenal. Post-renal failure occurs after obstruction of the urinary tract. In particular, the condition of the injury for electrocyte handling is totally different between pre-renal and renal causes. Furthermore, the cause of the post-renal AKI is totally different from the other two. Because the cause of post-renal AKI is the obstruction of the urinary tract.

Therefore, the authors should consider what kind of AKI is focused on in this study. Otherwise, the findings cannot be useful in the clinical setting.

3. The authors conducted an investigation on the association of important clinical factors with AKI development through a machine learning approach. The attempt to explore this research area is both significant and intriguing. Nevertheless, a critical concern arises from the ambiguity in defining the outcome. Additionally, the paper lacks a comprehensive discussion, which is essential for strengthening the overall analysis.

Reviewer 2 Report

Тhe proposed review paper is devoted to the description of a novel deep learning method for studying the role of important factors influencing the development of acute kidney injury and their interactions with each other. This task is important for assessment of the risk of acute kidney injury.

The authors describe previous results in the field and mention their advantages and disadvantages.

The proposed approach is described in detail. Information about the investigated groups of patients is provided. The available datasets are described. The machine learning models are presented. The obtained results are described and discussed. The authors conclude that the levels of serum creatinine, chloride, and magnesium are the three major risk factors that should be taken into consideration in the development of new clinal guidelines for assessing the risk of acute kidney injury.

The presentation of the main results is clear and comprehensive. The results are valuable and worthy of being published taking into account their possible development and applications in medicine, in particular for enhancement of the clinical practices, for investigation of the physiological mechanisms involved, development of early intervention and effective management strategies.

Minor revisions are suggested to improve the quality of the exposition:

p. 10, line 351: I suggest the number doi to be added to Ref. 5

p. 11, line 400: It should be “Therneau, T.; Atkinson, Ripley B.“ instead of “Therneau, T.; Atkinson, B. rpart“ in Ref. 28. 

Reviewer 3 Report

well written study. well done. It could be published without revision

Reviewer 4 Report

Overall it’s a good study .

There are few queries regarding the data presented

1. Is the table 2 depicts data at the time of ICU admission or first data available after ICU admission or the data at the time patient received in Emergency.

2. The values shown in Table 2 depict that mean Creatinine in the group who developed AKI had creatinine already above normal. (unless these values are those obtained after development of AKI).

3. How the authors diagnosed patients having prior renal disease?

4 If the decision tree has creatinine above 1.25 and that also in patient group who is old - it means already renal damage.

5. Line 257 “By focusing  on this specific group of patients, we were able to eliminate the confounding influences of  these conditions and examine the impacts of serum electrolyte levels on renal function in  a more controlled manner.” This means that the abnormal values of electrolytes is responsible for the AKI . Rather it should be that abnormality of these electrolytes  can be considered early biomarkers for the AKI. Or in other words the   early stage of pathophysiology of AKI may be affecting theses electrolytes whose abnormal values may indicate or be the biomarker of early stages of AKI.

Reviewer 5 Report

The abstract of the paper explains that assessing the risk of acute kidney injury (AKI) is a challenging issue for clinicians in intensive care units (ICU) and that the current clinical practice monitors only the level of serum creatinine and the volume of urine output. The authors aimed to develop machine learning models that not only illustrate how these factors interact with each other but also provide insights for developing new clinical practices to assess AKI risk. Below are my comments

1. The author did not introduce any technical novelty; instead, they utilized existing algorithms.

2. Could you please provide insight into the rationale behind employing both R and Python for the analysis?

3. Further elaboration is necessary regarding the chosen settings for the parameter ranges of DT and Random Forest models.

4. Typically, employing a higher number of folds in cross-validation enhances the accuracy estimation of the model's performance. It would be helpful to understand the reasons behind opting for 5-fold cross-validation instead of a higher number.

Round 2

Reviewer 1 Report

If the percentage of post-renal AKI is low, the result of the current study may be meaningful. So, I recommend that the authors check and add the percentage of post-renal AKI in patients treated in the ICU.

Reviewer 4 Report

Thank you for revising the manuscript as suggested.
